# ROCK2-induced metabolic rewiring in diabetic podocytopathy

Keiichiro Matoba [1✉], Yusuke Takeda[1], Yosuke Nagai[1], Kensuke Sekiguchi[1], Rikako Ukichi[1], Hiroshi Takahashi[1], Daisuke Aizawa[2], Masahiro Ikegami[2], Toshiaki Tachibana[3], Daiji Kawanami [4], Yasushi Kanazawa[1], Tamotsu Yokota[1], Kazunori Utsunomiya[5] & Rimei Nishimura[1]

Loss of podocytes is a common feature of diabetic renal injury and a key contributor to the development of albuminuria. We found that podocyte Rho associated coiled-coil containing protein kinase 2 (ROCK2) is activated in rodent models and patients with diabetes. Mice that lacked ROCK2 only in podocytes (PR2KO) were resistant to albuminuria, glomerular fibrosis, and podocyte loss in multiple animal models of diabetes (*i.e.*, streptozotocin injection, db/db, and high-fat diet feeding). RNA-sequencing of ROCK2-null podocytes provided initial evidence suggesting ROCK2 as a regulator of cellular metabolism. In particular, ROCK2 serves as a suppressor of peroxisome proliferator-activated receptors α (PPARα), which rewires cellular programs to negatively control the transcription of genes involved in fatty acid oxidation and consequently induce podocyte apoptosis. These data establish ROCK2 as a nodal regulator of podocyte energy homeostasis and suggest this signaling pathway as a promising target for the treatment of diabetic podocytopathy.

[1] Division of Diabetes, Metabolism and Endocrinology, Department of Internal Medicine, The Jikei University School of Medicine, Tokyo 105-8461, Japan. [2] Department of Pathology, The Jikei University School of Medicine, Tokyo 105-8461, Japan. [3] Core Research Facilities for Basic Science, Research Center for Medical Science, The Jikei University School of Medicine, Tokyo 105-8461, Japan. [4] Department of Endocrinology and Diabetes Mellitus, Fukuoka University School of Medicine, Fukuoka 814-0180, Japan. [5] Center for Preventive Medicine, The Jikei University School of Medicine, Tokyo 105-8461, Japan. ✉email: matoba@jikei.ac.jp

Chronic kidney disease (CKD) attributed to diabetes is a worldwide public health problem; it is the leading cause of end-stage renal disease and is associated with increased mortality from cardiovascular events. Persistent albuminuria is a hallmark of CKD among people with diabetes, which mainly arises from podocyte abnormalities. In both type 1 and type 2 diabetes, the number podocytes are decreased, and the reduced number of podocytes is the strongest predictor of the decline in the renal function[1,2]. Effective therapeutic strategies for the prevention and treatment of CKD are hampered by a poor understanding of the upstream abnormalities that lead to podocyte damage. In this context, a detailed analysis of the molecular mechanisms of podocyte injury is critically important.

Rho GTPase and its downstream effector Rho-associated coiled-coil containing protein kinase (ROCK), which regulate various cellular functions (e.g., contraction, motility, and proliferation) play important roles in the pathogenesis of systemic vascular diseases. Previous studies in our laboratory demonstrated that renal ROCK activity is elevated in animal models of both type 1 and type 2 diabetes[3]. Furthermore, ROCK inhibitor treatment could prevent diabetic albuminuria and glomerular fibrosis via regulation of the inflammatory process and hypoxic responses[4,5]. These experiments established the importance of ROCK as a therapeutic target for CKD in diabetes. ROCK has two isoforms; ROCK1 and ROCK2. The reno-protective actions of ROCK1 inhibition have been documented in a rodent model of diabetic renal injury[6]: on the other hand, the role of ROCK2 remains to be completely elucidated. These details are central towards a comprehensive understanding of ROCK biology in CKD. Cell-based studies have shown that ROCK2 mediates endothelial inflammation and mesangial damage by activating nuclear factor κB (NF-κB) signaling[7,8]. However, the specific functions of podocyte ROCK2 remain unclear.

In our current study, we identified a signaling pathway for the development of diabetic renal damage and suggest ROCK2 as an essential energy mediator in podocytes, thus advancing our knowledge of the molecular basis of CKD caused by diabetes. We found that Podocin-cre-mediated ROCK2 deletion in podocytes did not lead to any defects in the development or function of the kidney under physiological conditions; however, it alleviated albuminuria and glomerular sclerosis in three animal models of diabetes. Through genome-wide sequencing approaches, we demonstrated that ROCK2 modulates podocyte energy homeostasis and cell viability. These findings suggest that ROCK2 plays important roles in the development of diabetic podocytopathy and that targeting ROCK2-dependent signaling could serve as a promising strategy.

## Results

### Podocyte ROCK2 is upregulated in animal models of diabetes and in patients with diabetes

We first investigated the renal and glomerular-specific ROCK2 expression in rodent models of diabetes. As shown in Fig. 1a, ROCK2 was upregulated in the renal cortex of streptozotocin (STZ)-injected type 1 diabetic mice in comparison to mice treated with vehicle. The ROCK2 protein levels were also elevated in db/db mice and high-fat diet (HFD)-fed mice, both of these are common type 2 diabetes models for studies of diabetic kidney disease. In these models, RhoA (Supplementary Fig. 1a) and the downstream target myosin phosphatase targeting subunit 1 (MYPT1) are activated[3,9,10]. Moreover, there was a trend towards elevations in renal ROCK1 levels in diabetic animals (Supplementary Fig. 1b, c). Glomeruli isolated from these models exhibited a significant transcription-level increase in ROCK2 in comparison to glomeruli obtained from non-diabetic mice (Fig. 1b). Importantly, transcript datasets obtained from Nephroseq (https://www.nephroseq.org) showed a mild positive correlation between the glomerular expression of ROCK2 and urinary albumin excretion, measured as the albumin-to-creatinine ratio (ACR), in murine models of diabetes (Fig. 1c), that may suggest the association of activated ROCK2-signaling with albuminuria. Double immunostaining of the glomeruli with antibodies against ROCK2 and Nephrin confirmed their co-localization, indicating that ROCK2 is expressed in podocytes (Fig. 1d). Importantly, podocyte ROCK2 levels were increased in mice following STZ-injection, db/db mice, and by HFD treatment. A similar activation pattern of podocyte ROCK2 was observed in human kidney tissue sections (Fig. 1e). Taken together, these data indicate the association of ROCK2 with diabetic podocyte injury.

### Podocyte-specific genetic deletion of ROCK2 does not cause albuminuria in mice

To determine the role of podocyte ROCK2 in vivo, we generated podocyte-specific ROCK2 knockout (hereafter referred to as PR2KO) mice by breeding ROCK2[flox/flox] mice with mice expressing cre recombinase under the control of Podocin promoter (Fig. 2a–d). These mice displayed no significant differences in body appearance, kidney size (Fig. 2e), kidney histology, or podocyte morphology (Fig. 2f) in comparison to wild-type (WT) mice. Observations of body weight and ACR revealed no abnormalities in the body growth (Fig. 2g) or kidney function (Fig. 2h) of PR2KO mice, even at 12 months of age.

### Ablation of podocyte ROCK2 prevents diabetic renal damage in mice

To investigate whether the deletion of ROCK2 in podocytes rescues some of the key features of CKD in diabetes, we injected STZ, crossed with db/m mice to generate db/db, or challenged mice with HFD (Fig. 3a). Of note, the selective deletion of ROCK2 in podocytes resulted in significant protection against diabetes-induced albuminuria (Fig. 3b) and renal hypertrophy (Fig. 3c), without affecting body weight (Supplementary Fig. 2a) or blood glucose levels (Supplementary Fig. 2b). In comparison to ROCK2-floxed littermates, those lacking ROCK2 exclusively in podocytes were completely impervious to mesangial expansion (Fig. 3d). While significant disruption in Wilms tumor 1 (WT1) was observed in diabetic mice, PR2KO mice were protected from podocytes loss (Fig. 3e). TUNEL-assay showed that diabetes-induced glomerular cell apoptosis was decreased in PR2KO mice (Supplementary Fig. 2c). As shown in Fig. 3f, g, transmission electron microscopy (TEM) micrographs from PR2KO mice revealed a significant reduction in foot process width and glomerular basement membrane (GBM) thickness in comparison to ROCK2-floxed mice in the setting of diabetes. Hence, in three different methodologies using pharmacologically, genetically, and diet-induced diabetes, the deletion of ROCK2 prevented podocyte damage, and as a consequence, the glomerular morphology was preserved.

### ROCK2 deletion results in robust metabolic signature in non-stimulated podocytes

We then explored the molecular mechanisms by which ROCK2 deficiency exerts protective effects in podocytes. We deleted ROCK2 in immortalized podocytes and characterized the transcriptomic profiles by performing unbiased RNA-sequencing (Fig. 4a and Supplementary Fig. 3a, b). An analysis of all expressed genes revealed 298 significantly downregulated and 306 significantly upregulated genes (fold change > 1.5; p value < 0.05) (Fig. 4b, c). As shown in Fig. 4d, a KEGG pathway enrichment analysis revealed significant changes in the metabolic pathway. Upregulation of core essential genes in fatty acid metabolism, including Cpt1b, Cpt2, Acox2, Ppara, were

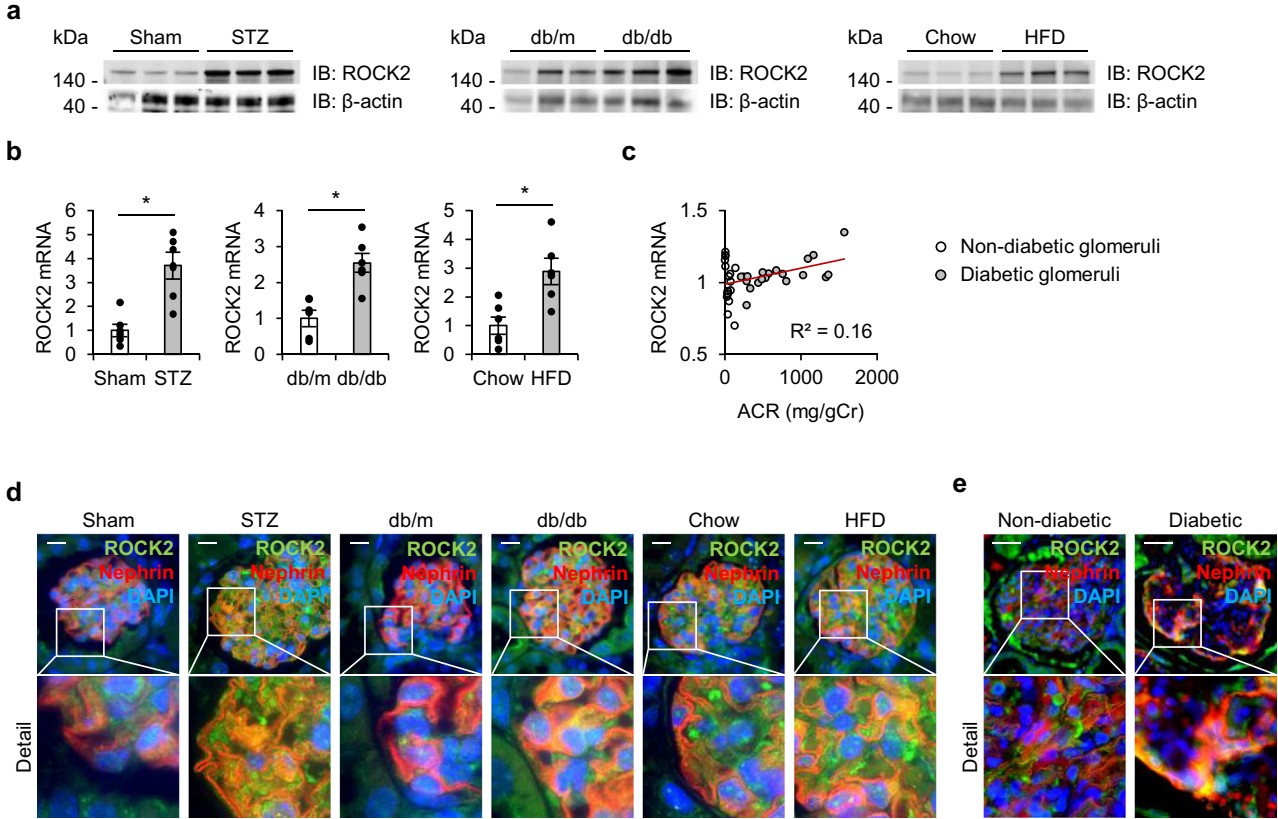

**Fig. 1 Podocyte ROCK2 is upregulated in animal models of diabetes and in patients with diabetes. a** ROCK2 levels in the renal cortex of streptozotocin (STZ)-injected mice, db/db mice, and mice treated with high-fat diet (HFD). **b** The expression levels of ROCK2 in isolated glomeruli. **c** A correlation analysis between glomerular ROCK2 mRNA and albumin-to-creatinine ratio (ACR). Hodgin Diabetes Mouse Glom data obtained from the transcriptomic database Nephroseq (https://www.nephroseq.com) were used. The trendline is shown in red. The R value was determined by Pearson correlation analysis; white and gray plots represent non-diabetic mice and diabetic mice, respectively (n = 18–21). **d** Immunostaining of ROCK2 (green) and Nephrin (red) in the glomerulus of STZ-injected mice, db/db mice, and HFD-fed mice. Nuclei were visualized with DAPI (blue). The scale bar on the top left represents 10 μm. **e** Immunostaining of ROCK2 (green) and Nephrin (red) in the glomeruli of patients without or with diabetes. Nuclei were visualized with DAPI (blue). The scale bar on the top left represents 50 μm. *$p < 0.05$. Data represent the mean ± s.e.m.

confirmed by qPCR analysis in podocytes treated with ROCK2 siRNA (Supplemental Fig. 3c) or ROCK2 inhibitor (Supplemental Fig. 3d). Among these genes, Cpt1b, Acox2 (Supplementary Fig. 3e), and Ppara (Fig. 5i) were significantly upregulated in glomeruli isolated from PR2KO mice compared to WT mice. These data indicate that ROCK2 is involved in essential pathway of fatty acid oxidation (FAO) in podocytes.

**Loss of ROCK2 improves fatty acid metabolism.** Among renal cells, podocytes are considered to be particularly sensitive to fatty acid accumulation[11,12]. Data from relevant studies indicate that increased FAO is a protective adaptive response that helps podocytes to deal with excess fatty acid. This notion is supported by a previous report demonstrating that a single-nucleotide polymorphism in FAO-related genes is strongly associated with proteinuria[13,14], and that key proteins involved in FAO, including PPARα and CPT1, are downregulated in renal tissue obtained from patients with coexisting diabetes and CKD[15]. In podocytes, the pharmacological arrest of FAO by etomoxir leads to a loss of ATP production (Fig. 5a), increased apoptotic marker levels (Fig. 5b) and TUNEL-positive cells (Fig. 5c), indicating that FAO is essential in podocyte energy homeostasis. Transforming growth factor β (TGF-β) drives glomerular damage in diabetes[16], and this is upregulated in glomeruli isolated from diabetic mouse models (Fig. 5d). TGF-β significantly attenuates FAO in podocytes;

however, the genetic inhibition of ROCK2 reverses the utilization of fatty acid (Fig. 5e). Consistently, the downregulation of FAO enzymes (i.e., Fatp1, Cpt1a, Acox1) were prevented in these settings (Fig. 5f), indicating that ROCK2 serves as negative regulator of energy production and cellular homeostasis. We reasoned that one possible explanation for this action could be through the effect of ROCK2 on PPARα, since the key FAO enzymes were upregulated by PPARα agonist in podocytes (Supplemental Fig. 4a) and the genetic ablation of ROCK2 was associated with recovery from the PPARα suppression induced by TGF-β (Fig. 5g). As demonstrated in Fig. 5h, TGF-β-induced podocyte apoptosis was inhibited in the setting of ROCK2 deletion. This beneficial action was partially canceled by the treatment with etomoxir, indicating that cytoprotective effects of ROCK2 inhibition is dependent, at least in part, on fatty acid oxidation. Treatment with fenofibrate showed similar protective effect as ROCK2 deletion in podocytes, and co-treatment with fenofibrate and ROCK2 siRNA induced additive effects on podocyte protection. These data support the idea that the beneficial actions of ROCK2 deficiency is dependent on PPARα activation. The upregulation of PPARα and FAO, and the prevention of podocyte apoptosis were also observed with the pharmacological blockade of ROCK2 (Supplemental Fig. 4b–e). The expression levels of TGF-β were not changed in ROCK2-dificient conditions (Supplemental Fig. 4f) and in podocytes treated with ROCK2 inhibitor (Supplemental Fig. 4g). Consistent with the findings

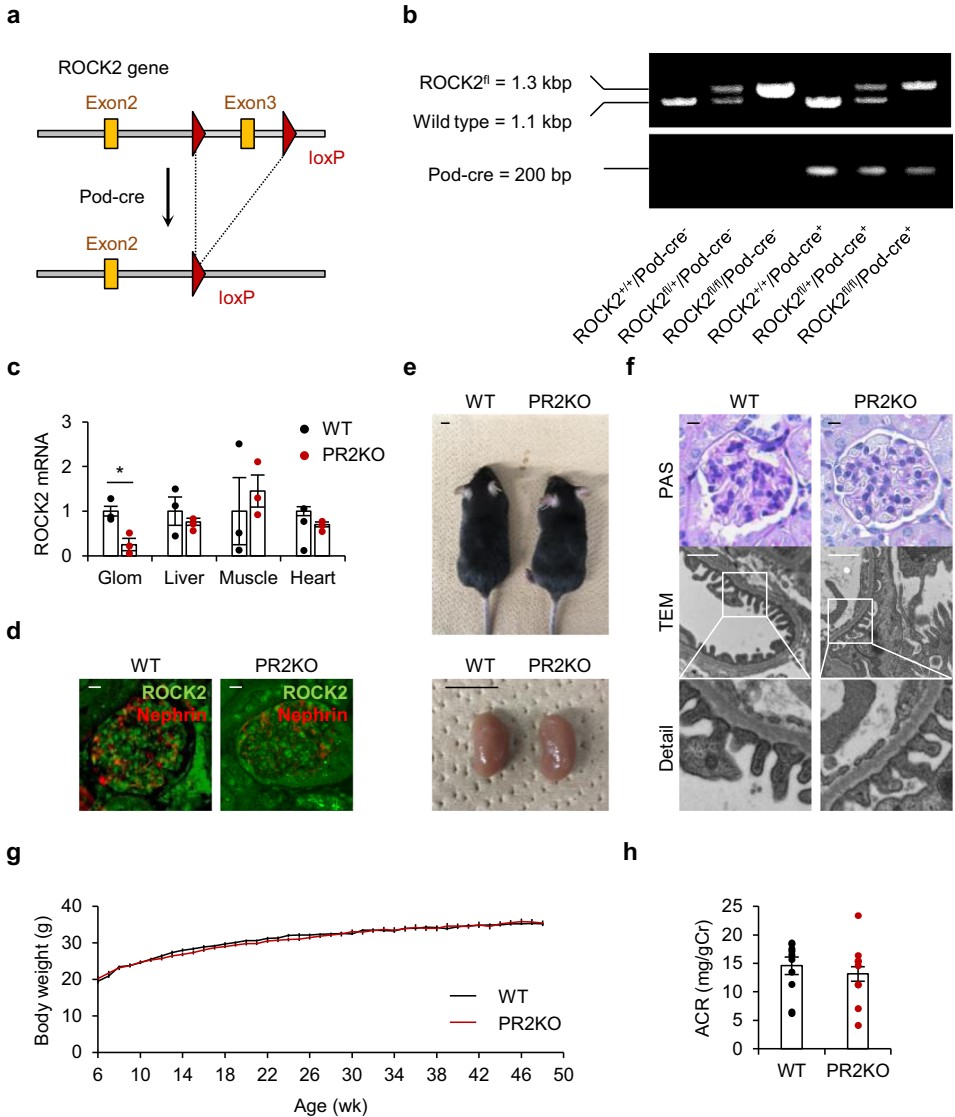

**Fig. 2 Generation of podocyte-specific ROCK2-deficient mice. a** The strategy for the generation of podocyte-specific ROCK2 knockout mice. Gene-targeting vectors were constructed to delete exon 3 of ROCK2. **b** Representative genotyping PCR detecting transgenic ROCK2-floxed alleles and Podocin-Cre (Pod-Cre). **c** Confirmation of tissue-restricted-ROCK2 deletion in podocyte-specific ROCK2 knockout mice (PR2KO). The mRNA levels of ROCK2 in the glomerulus (Glom), liver, muscle, and heart are demonstrated ($n = 3$). **d** Immunolabeling of ROCK2 and the expression of nephrin in kidney sections obtained from WT and PR2KO mice. The scale bar at the left top represents 10 μm. **e** Representative images of the whole body (upper panel) and kidney (lower panel) in each genotype. The scale bar on the top left represents 1 cm. **f** Representative PAS staining and transmission electron microscope (TEM) images of the glomerulus from WT and PR2KO mice. The scale bars at the top represents 10 μm and 1 μm, respectively. **g** Body weight information in aged WT and PR2KO ($n = 9–11$). **h** ACR at 12 months of age ($n = 9–10$). $*p < 0.05$. Data represent the mean ± s.e.m.

obtained from in vitro experiments, glomerular PPARα was upregulated in PR2KO mice (Fig. 5i) and there was a mild negative correlation between glomerular ROCK2 and the expression of PPARα (Fig. 5j), further indicating a role of ROCK2 as a potential negative regulator of PPARα. An immunoprecipitation assay revealed that ROCK2 does not bind to PPARα in podocytes (Supplemental Fig. 4h), suggesting an indirect action of ROCK2 on PPARα.

## Discussion
Our finding that podocyte-ROCK2 knockout mice are protected from diabetes-induced renal damage is somewhat surprising in light of the appreciation that small GTPases (i.e., RhoA, Rac1, Cdc42) are critical for actin cytoskeleton reorganization and the

regulation of cell structure[17]. In this context, one important consideration is the issue of GTPase specificity. Several studies have illustrated the role of small GTPases by analyzing mice with podocyte-intrinsic gene manipulations. It is noteworthy that the cell division cycle 42 (Cdc42) conditional knockout model develops severe proteinuria and dies at around 3 weeks of age[18]. However, mice that only lacked RhoA or Rac family small GTPase 1 (Rac1) in podocytes developed no phenotype. From the therapeutic perspective, these observations indicate that interventions targeting the RhoA-ROCK2 axis could be a safe and feasible approach to prevent the progression of CKD.

Mechanistic studies exploring the ROCK2-PPARα axis are in agreement with a recent observation demonstrating that the inhibition of Rho/ROCK signaling enhances PPARα ligands in oligodendrocytes to protect against cytokine toxicity[19]. However,

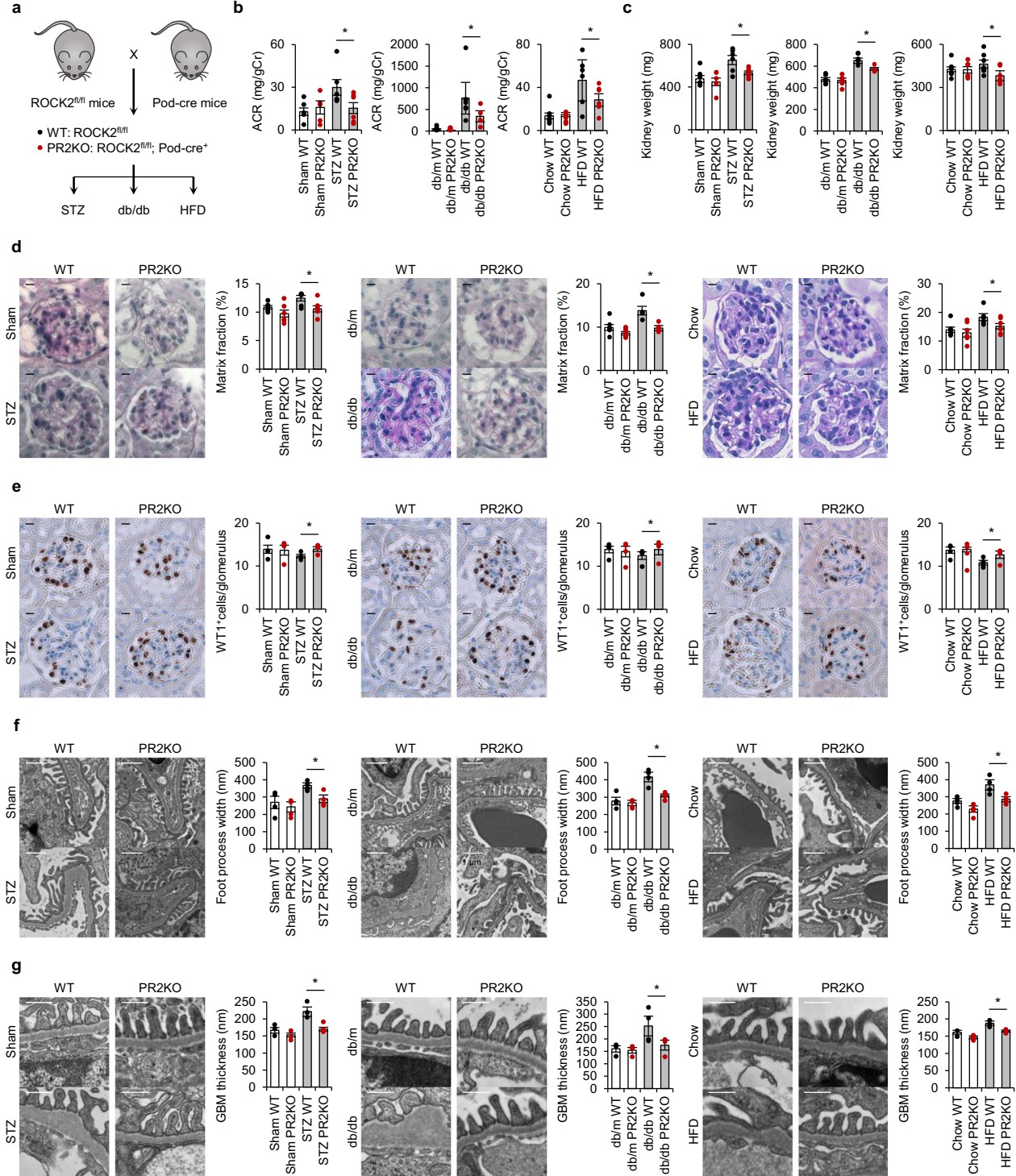

**Fig. 3 Ablation of podocyte ROCK2 prevents diabetic renal damage in mice. a** The breeding scheme that was used to generate PR2KO mice. Diabetes was induced by STZ injection, mating with db/m mice to generate db/db mice, or HFD feeding. **b** ACR in STZ-injected, db/db, HFD-fed WT and PR2KO mice ($n = 4$–8). **c** The kidney weight in the three experimental groups ($n = 4$–8). **d** Representative PAS-stained images of kidney glomeruli from mice. The scale bar on the top left represents 10 μm ($n = 4$–6). **e** Wilms tumor 1 (WT1) immunostaining and quantification of WT1-positive cells in glomeruli from mice. The scale bar on the top left represents 10 μm ($n = 4$). **f** Foot process width examined by transmission electron microscopy. The scale bar on the top left represents 1 μm ($n = 4$). **g** Glomerular basement membrane (GBM) thickness assessed by transmission electron microscopy. The scale bar on the top left represents 0.5 μm ($n = 4$). *$p < 0.05$. Data represent the mean ± s.e.m.

the molecular mechanism by which ROCK2 regulates the expression of PPARα remains incompletely understood. Considering the broad essential roles of ROCK2 signaling in cellular biology, as revealed by RNA-sequencing, we cannot exclude the

possibility that the induction of PPARα in ROCK2-null podocytes could be a result of another cellular event.

In line with our observations, previous studies have demonstrated protective actions of PPARα agonists in cultured

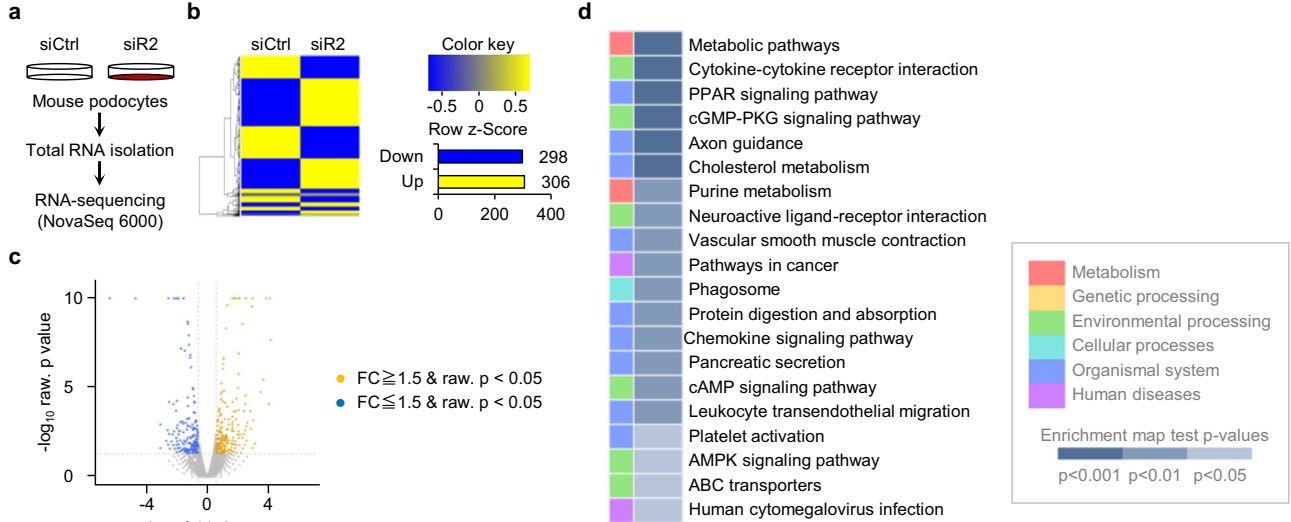

**Fig. 4 RNA-sequencing reveals the metabolic signature in ROCK2-null podocytes. a** The experimental approach to generate samples for high throughput RNA-sequencing. Murine podocytes were treated with negative control (siCtrl) or siRNA against ROCK2 (siR2). **b, c** A heat map (**b**) and volcano plot (**c**) of differentially expressed genes in podocytes based on log (fold change) >1.5 with adjusted $p < 0.05$. **d** Results of a gene-set enrichment analysis ($p$ value top 20).

podocytes[20]. Moreover, a link between PPARα and diabetic renal disease became apparent after two prospective clinical trials, the Fenofibrate Intervention and Event Lowering in Diabetes (FIELD) study and the Action to Control Cardiovascular Risk in Diabetes (ACCORD) study[21,22]. These studies demonstrated that fibrate, a PPARα agonist, was effective for improving albuminuria in patients with type 2 diabetes. When taken into the context of our current data, it is plausible to conclude that the beneficial renal actions of ROCK2 inhibition in diabetes may be due to the re-sensitization of podocytes to insulin signaling or prevention of metabolic stress in association with the recovery of FAO. However, the role of dysregulated fatty acid metabolism in diabetic renal fibrosis is only beginning to be elucidated. Further studies are needed to understand the association between metabolic abnormalities in podocytes and the process towards end-stage renal disease. In addition, the forced induction of genes involved in FAO by the overexpression of PPARγ coactivator 1α (PGC1α) results in collapsing glomerulopathy[23]. Increased reactive oxygen species derived from excess FAO is also reported[24]. These data indicate that a delicate and precise balance of fatty acid metabolism is critical for the function of podocytes and there is a narrow therapeutic window.

When considered alongside previous findings demonstrating the pathogenic roles of ROCK2 in endothelial injury and mesangial damage, the current observations revealed that ROCK2 as a central regulator in glomerular pathological process. We note that ROCK1 has also been shown to promote podocyte damage via the regulation of mitochondrial fission[6]. In addition, ROCK1 induces endothelial-to-mesenchymal transition in glomeruli to aggravate albuminuria in diabetes[25]. Thus, ROCK1 could be another therapeutic target for preventing podocyte and endothelial damage. Studies to elucidate the role of ROCK isoforms in each renal cell type will provide additional insights into its pathologic role.

As summarized in Fig. 6, the present study identifies podocyte ROCK2 as a crucial integrator of podocytopathy, which exacerbates glomerular damage. We suggest that the beneficial effects of ROCK2 inhibition are mediated by the recovery of podocyte fatty acid metabolism, secondary to the rescue of PPARα signaling. When the upregulation of podocyte ROCK2 in subjects with

diabetes is taken into consideration, targeting this machinery might provide meaningful renal benefits in humans.

## Methods

**Mice.** The floxed-ROCK2 mouse line (C57BL/6 background) was generated by Transgenic by inserting the LoxP site flanking exon 3 of the Rock2 gene. Podocyte-specific ROCK2 knockout mice (PR2KO) were generated by crossing ROCK2^flox/flox mice and NPHS2 (also known as Podocin)-cre mice obtained from The Jackson Laboratory. Genotyping was performed using tail DNA at the timing of weaning. PCR primers included: ROCK2 flox forward: 5′-AAAGAAGTGGGTTAAGGAT CATTGC-3′, ROCK2 flox reverse: 5′-TGCTTGGATTAAAGATATTCACCGAC AAG-3′, cre forward: 5′-GCGCTGCTGCTCCAG-3′, cre reverse: 5′-CGGTTATTC AACTTGCACCA-3′. The WT allele was detected as a band at 1.1 kbp, whereas the floxed allele was detected as a band at 1.3 kbp. We used ROCK2-floxed, cre-negative littermates as our control. Mice were fed a standard chow unless otherwise specified, kept on a daily 12 h light/dark schedule, and monitored for signs of ill health every other day. Only male mice were used for studies.

For the STZ experiment, 8-week-old mice were intraperitoneally injected with STZ (50 mg/kg/day) or vehicle (Na-Citrate) on 5 consecutive days to induce diabetes. Diabetic db/db mice and their non-diabetic littermates, db/m, were obtained from The Jackson Laboratory. Since db/db mice are infertile, db/db PR2KO mice were bred by mating db/m ROCK2^flox/flox mice with db/m PR2KO mice. Lean controls were db/m ROCK2^flox/flox animals. For the high-fat diet experiment, 6-week-old mice were fed either standard chow or a diet containing 41% kcal from fat for 12 weeks. Body weight was monitored weekly. At the end of the study, serum and urine were collected. Renal tissues were snap-frozen in liquid nitrogen for a biochemical analysis, or fixed in 10% neutral buffered formalin for histological assessment. All experiments involving animals were conducted under protocols approved by the Committee on Ethical Animal Care and Use of The Jikei University School of Medicine.

**Glomerular isolation.** Glomeruli were isolated as described previously[7]. Briefly, mice were perfused with magnetic Dynabeads. The kidneys were minced into small pieces, and digested by collagenase A in Hank's balanced salt solution. The digested kidney was then filtered through 100-μm and 70-μm cell strainer. Glomeruli containing Dynabeads were collected using a magnet.

**Cell culture.** No cell lines used in this study were found in the database of commonly misidentified cell lines that is maintained by ICLAC and NCBI BioSample. A conditionally immortalized murine podocyte cell line (E11) was obtained from Cell Line Services, but was not further authenticated after purchase. Podocytes were propagated at 33 °C in RPMI1640 medium supplemented with 10% fetal bovine serum and 10 U/mL of interferon-γ to enhance the expression of thermosensitive T antigen. In order to induce differentiation, podocytes were maintained at 37 °C without interferon-γ for 10–14 days. ROCK2 knockdown podocytes were established by incubating cells with transfection mixtures containing 50 nM of siRNA. A scrambled sequence that will not lead to the specific degradation was used as

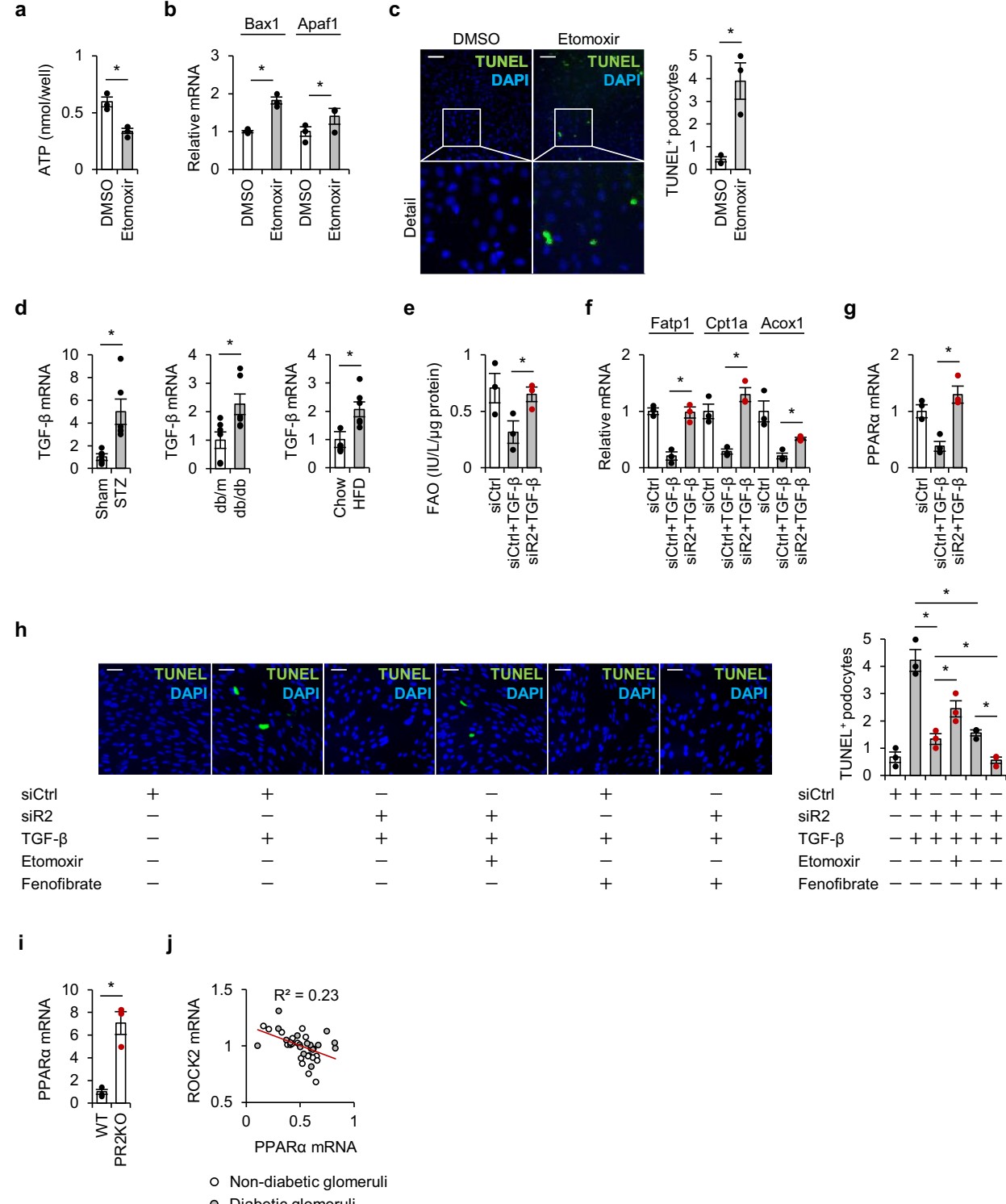

control. Transfection of siRNAs with 1 μL of Lipofectamine reagent (Invitrogen) for 6 h resulted in ~63% mRNA knockdown and 71% knockdown at protein levels (Supplemental Fig. 3a, b). Twenty-four hours after adding growth medium, the transfected podocytes were stimulated with TGF-β (5 ng/mL) for 24 h.

**Histology**. Formalin-fixed paraffin-embedded human specimens were obtained from the ProteoGenex and stored at 4 °C until use. All donors have voluntarily signed legal Informed Consent documents, which clearly state the intent of use for their donated specimens, and kidney samples were obtained following official protocols, with appropriate Institutional Review Board/Independent Ethics Committee (IRB/IEC) approval. In total, 8 postmortem kidney slices were examined, including 4 control kidney slices from healthy donors and 4 from patients with

diabetes and CKD. The patients' age at the time of death, sex, body mass index, postmortem interval, pathological diagnosis, cause of death, and comorbidities are demonstrated in Supplemental Table 1. Representative images were obtained from normal patient #1 and diabetic nephropathy patient #1. Mice were transcardially perfused with PBS. Tissues were post-fixed in 10% neutral buffered formalin, embedded in paraffin, and cut into 3-μm-thick sections. Twenty glomeruli were quantified per section using the ImageJ software program (https://imagej.nih.gov/ij). The mesangial matrix fraction was calculated as the area of PAS-positive staining per total glomerular tuft area. For immunofluorescence, 4-μm-thick paraffin-embedded sections were deparaffinized and subjected to antigen retrieval in citrate buffer. The following antibodies were used for staining: anti-ROCK2 (abcam #ab71598, dilution 1:200), anti-Nephrin (abcam #ab227806, dilution 1:2000;

**Fig. 5 ROCK2 deletion improves fatty acid metabolism. a–c** The amounts of ATP (**a**), the expression of markers of apoptosis (**b**), the number of TUNEL-positive podocytes (**c**) treated with vehicle (DMSO) or etomoxir for 24 h. The scale bar on the top left represents 100 μm ($n = 3$). **d** The glomerular expression of TGF-β in STZ-injected mice, db/db mice, and mice treated with HFD ($n = 6$). **e** FAO was assessed using cell lysates obtained from podocytes treated with siRNA against ROCK2 ($n = 3$). **f** Relative mRNA levels of FAO mediators in podocytes treated with siRNA against ROCK2 before stimulation with TGF-β ($n = 3$). **g** PPARα mRNA levels in podocytes treated with siRNA against ROCK2 before stimulation with TGF-β ($n = 3$). **h** Representative microphotographs and the quantification of TUNEL-positive apoptotic podocytes. Podocytes were pretreated with siRNA against ROCK2 before stimulation with TGF-β. Etomoxir was used in order to inhibit fatty acid oxidation in podocytes treated with ROCK2 siRNA. Fenofibrate was used to investigate the effect of PPARα activation on podocyte apoptosis. The scale bar on the top left represents 50 μm ($n = 3$). **i** Glomerular PPARα mRNA levels in WT and PR2KO mice. **j** A correlation analysis between glomerular transcripts of ROCK2 and PPARα. Hodgin Diabetes Mouse Glom data obtained from the transcriptomic database Nephroseq (https://www.nephroseq.com) were used. The trendline is shown in red. The R value was determined by Pearson correlation analysis; white and gray plots represent non-diabetic mice and diabetic mice, respectively ($n = 18–21$). *$p < 0.05$. Data represent the mean ± s.e.m.

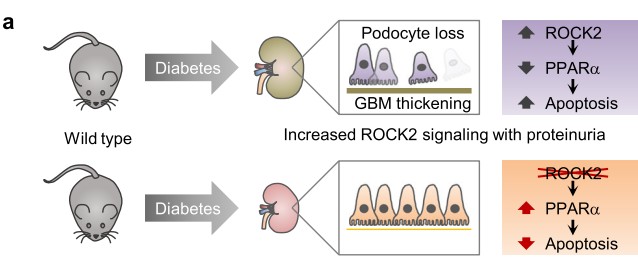

**a**

Wild type — Diabetes → Increased ROCK2 signaling with proteinuria

Podocyte loss / GBM thickening / ↑ ROCK2 ↓ PPARα ↓ ↑ Apoptosis

Podocyte ROCK2 KO — Diabetes → Improved PPARα-regulated metabolism

~~ROCK2~~ ↑ PPARα ↓ ↓ Apoptosis

**Fig. 6 ROCK2 inhibition attenuates diabetic podocytopathy through targeting PPARα. a** A schematic summary of key observations. In diabetic kidneys, podocyte ROCK2 is upregulated with proteinuria. Podocyte-specific ROCK2 deletion prevents the loss of podocytes by recovering PPARα signaling.

ORIGENE #1243-1256, dilution 1:25), and anti-WT1 (abcam #ab89901, 1:300). For double immunostaining with ROCK2 and nephrin in mice, 20 glomeruli were observed per section and representative images taken at the same depth are demonstrated. To determine the loss of glomerular podocytes, the number of WT1-positive cells was counted in 20 glomeruli per section.

**Transmission electron microscopy**. For electron microscopy, the specimens were fixed with 2% glutaraldehyde in 0.1 M phosphate buffer overnight at 4 °C and processed at the Electron Microscopy Facility at The Jikei University School of Medicine. The samples were then post-fixed with 1% osmium tetroxide in the same buffer at 4 °C for 2 h. Dehydration was performed using a graded ethanol series, then the specimen was placed in propylene oxide, and embedded in Epok 812 (Oken). Ultrathin sections were prepared with a diamond knife, and then stained with uranium acetate and lead citrate. The sections were examined by a pathologist using a JEM-1400 Plus transmission electron microscope (JEOL) at 100 kV. The GBM thickness was determined as the distance between the outer limit of the endothelium and the cell membrane of the podocyte foot process. Twenty positions of each mouse were measured using the straight-line tool of ImageJ.

**RNA isolation, quantitative real-time PCR, and RNA-sequencing**. Total RNA was isolated from podocytes with TRIzol reagent (Invitrogen) followed by chloroform-isopropanol extraction and ethanol precipitation, and 1 μg of total RNA was reverse-transcribed using the iScript RT reagent Kit (Bio Rad). To evaluate the mRNA expression, real-time quantitative PCR was performed using a Thermal Cycler Dice Real Time System TP800 (Takara Bio) with SYBR Green I fluorescence signals. The mRNA levels were normalized to β-actin and expressed as levels relative to control. The primer sequences used for amplifications are presented in Supplemental Table 2. RNA-sequencing was performed by Macrogen using the Illumina NovaSeq6000 platform (Illumina).

**Western blotting**. Renal tissues or podocytes were lysed with RIPA buffer containing a protease inhibitor cocktail. Equal amounts of protein samples were subjected to western blotting as described previously[3]. Immunoreactive bands were visualized with an enhanced chemiluminescence system (Amersham, GE Healthcare Life Science). The peroxidase luminescence intensity was measured using LAS-4000 mini Luminescent Image Analyzer (FUJIFILM). The primary antibodies used were as follows: anti-ROCK2 (#ab71598, dilution 1:1000) was from abcam; anti-ROCK1 (#sc-17794, dilution 1:1000) was from Santa Cruz; and anti-PPARα (#PA1-822A, dilution 1:1000) was obtained from Thermo Fisher Scientific.

**RhoA activity assay**. Renal RhoA activity was quantified using G-LISA RhoA activation kit (Cytoskeleton) according to the manufacturer's protocols. Briefly, cortical protein samples were added to microplate coated with Rho-GTP binding protein. The plates were then incubated with RhoA antibody and secondary horseradish peroxidase–conjugated antibody. The luminescence signal was detected by a microplate spectrophotometer.

**Fatty acid oxidation assay**. Fatty acid oxidation in cultured podocytes was assessed using a fatty acid oxidation assay kit (Biomedical research service center, University of Buffalo, State University of New York) according to manufacturer's instructions.

**TUNEL assay**. Apoptotic podocytes were visualized by the terminal deoxynucleotidyl transferase-mediated deoxyuridine triphosphate nick end labeling method using an in-situ apoptosis detection kit (Takara Bio), in accordance with the manufacturer's protocol. In order to evaluate apoptotic cells in glomeruli, 20 glomeruli were examined per section.

**Statistics and reproducibility**. Data are represented as the mean ± s.e.m. ($n$ as indicated in the figure legends). The measurements were taken from distinct samples. Statistical evaluations of two groups were performed using a two-tailed Student's $t$ test, with the exception of the urinary dipstick test data, which were compared using the Mann–Whitney $U$ test. Data involving more than two groups were assessed using an ANOVA followed by Bonferroni's post hoc correction. Pearson's correlation was applied to analyze the associations between ROCK2 mRNA levels and urinary albumin, and PPARα transcript levels. $P$ values of <0.05 were considered to indicate statistical significance.

**Reporting summary**. Further information on research design is available in the Nature Research Reporting Summary linked to this article.

## Data availability

The source data underlying the graphs and charts in the main manuscript file are shown in Supplementary Data 1. RNA-sequencing data from this study have been deposited online and available at https://figshare.com/articles/dataset/RNA-Seq_xlsx/19328630.

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

## Acknowledgements

We gratefully acknowledge Yuko Niikura, Yuki Takemura, Kazuya Sakurai, and Mamiko Owada for their technical assistance. The authors also thank Shigeru Kageyama for his advice on the clinical analysis. K.M. and D.K. are supported by JSPS KAKENHI (Grant Number 20K08645, 19K08714, and 18K15985). K.M. is also supported by Suzuken Memorial Foundation, Takeda Science Foundation, Yokoyama Foundation for Clinical Pharmacology, MSD Life Science Foundation, Ichiro Kanehara Foundation, Japan Diabetes Foundation, and Uehara Memorial Foundation.

## Author contributions

K.M. designed and performed research, analyzed data, and wrote the manuscript. Y.T., Y.N., K.S., and D.K. assisted in the in vivo experiments. R.U. and H.T. helped with the statistical analysis. D.A., M.I, and T.T. helped with the pathological analysis. Y.K., T.Y., K.U., and R.N. wrote the manuscript. All authors read and commented on the manuscripts.

## Competing interests

The authors declare no competing interests.
