## [Peer Review File · Communications Biology]

Reviewers' comments:

Reviewer #1 (Remarks to the Author):

In this study, the researchers investigated the function of ROCK2 in kidney disease using three disease models. The aim and importance of study is clear, but the manuscript has flaws for a publication. It seems that more comprehensive data is needed overall to obtain the decisive conclusion authors suggest. Followings are some points that should be addressed.

Authors showed the changes in ROCK2 mRNA and protein level in three kinds of disease model in figure 1. What happen with ROCK-related signals such as RhoA, ROCK1, MYPT and so on?

In figure 1c and 5j, isn't R2 low to say that two variants are closely related each other?

In figure 1d, the image seems not well focused. It seemed that the images were taken at different depth when considering the morphology of nucleus. Depending on the depth, the brightness will be measured differently. Also, how many fields were taken and tested?

How many animals and images were examined in figure 3? To examine the significance, more tests will be needed.

In RNA-sequencing analysis, supportive experiments are needed, for example qRT-PCR analysis for fatty acid metabolism-related genes.

Authors showed the effect of ROCK2-deficiency on TGF β in podocyte in Figure 5. It seems that ROCK2 downregulate TGF β in the results. What is the link between ROCK2 and TGF β expression?

Also, what is the effects of ROCK2 deficiency on fatty acid metabolism in cells and animals?

Was podocyte cell death detected in disease models, and the recovery from ROCK2-deficient mice?

Reviewer #2 (Remarks to the Author):

In the manuscript entitled "ROCK2-induced metabolic rewiring in diabetic podocytopahty" submitted by Matoba et al, the authors investigated the role of podocyte ROCK2 in response to diabetic injury. They reported that podocyte ROCK2 is activated in 3 different rodent diabetic models and in patients with diabetes. They found that selective deletion of ROCK2 in podocytes has no effect on development but protects against diabetic kidney injury in different diabetic models. They further found that the beneficial effect of ROCK2 inhibition is due to rescue of PPAR α signaling, leading to recovery of podocyte fatty acid metabolism. This is a well-designed and well-performed study with clear results and presentation. A minor issue is that high resolution double IF photos are needed.

Reviewer #3 (Remarks to the Author):

In this report, Matoba and colleagues demonstrate a novel role for ROCK2 in podocytes in mediating diabetic renal complications. They do so using a combination of a novel transgenic animal model lacking ROCK2 in podocytes (PR2KO) and immortalized podocytes treated in culture with anti ROCK2 siRNA. Inhibition of ROCK2 limits renal injury in three different models of diabetes. They also demonstrate a role for ROCK2 in modulating PPAR α expression and fatty acid oxidation in podocytes. Overall, the studies are well-described and carried out, but certain conclusions may not be fully supported or could be further strengthened.

Specific comments:

Methods: For db/db model, why were db/+ ROCK2 $^{flox/flox}$ mice used as controls rather than db/db ROCK2 $^{flox/flox}$? db/+ are not suitable controls in this scenario.

Figure 4 - what level of ROCK2 inhibition was achieved? What does siCtrl refer to - a control gene or a scrambled control? How much lipofectamine was used and for how long?

In Figure 4, transcriptomics were carried out in immortalized podocytes rather than in primary cells. While this is experimentally justified, this information should be more clearly conveyed in the text accompanying Figure 4 (i.e. line 111, not only in the methods section). Further, it should be clarified in the text that transcriptional differences observed are following stimulation by TGF-b,

and the rationale for the use of TGF- β should be clarified here, rather than later in the manuscript.

On a related note, was a baseline (non-TGF β treated) transcriptomics experiment carried out?

Fig. 4B: Label as siR2+TGF β , as in Fig. 5g.

In Figure 5F, while authors show that treatment of podocytes with siR2 prevents apoptosis, they do not demonstrate that this is dependent on fatty acid oxidation. In other words, if these cells were also treated with etomoxir, would the protective effect of siR2 be lost? Also, if these effects in FAO are mediated by PPAR α , does treatment with fibrates induce the same protective effect as ROCK2 inhibition? And would co-treatment with siR2 and fibrates result in additive or no additive effects? These crucial experiments are necessary to conclude that the cytoprotective effects of ROCK2 inhibition are due to enhanced FAO and that this enhancement of FAO is due to PPAR α activation. This is especially true given the reported changes in AMPK signaling (Fig. 4), another mechanism through which FAO may be regulated.

As shown here, the data demonstrate that ROCK2 inhibition is protective in the context of diabetes-induced renal injury and that ROCK2 inhibition is ASSOCIATED with rescue of FAO and PPAR α expression. However, the data as shown do not support the conclusion that the cytoprotective effects of ROCK2 inhibition are mediated by PPAR α or by induction of FAO. These endpoints could just as easily be a consequence of other changes occurring due to ROCK2 inhibition.

Responses to reviewers' comments

We would like to express our appreciation editors for their comments and support of this work. Reviewer's comments are given in italics. Please find below a point-by-point response to their comments and suggestions.

***Reviewer #1:** In this study, the researchers investigated the function of ROCK2 in kidney disease using three disease models. The aim and importance of study is clear, but the manuscript has flaws for a publication. It seems that more comprehensive data is needed overall to obtain the decisive conclusion authors suggest. Followings are some points that should be addressed.*

Our response - We would like to thank the reviewer for kind review and positive statements and for the additional queries below that have helped strengthen the manuscript.

Comment 1 - Authors showed the changes in ROCK2 mRNA and protein level in three kinds of disease model in figure 1. What happen with ROCK-related signals such as RhoA, ROCK1, MYPT and so on?

Our response – Thank you for your constructive comments. In the additional experiments, we have investigated the ROCK-related signals in murine models of diabetes. As shown in Supplementary Fig. 1a, renal RhoA activity was elevated in three kinds of diabetes models. Consistently, it has been already reported that phosphorylated form of MYPT1 is increased in the kidneys of STZ-injected models (Mol Med Rep. 2015; 12: 45, PMID: 25695625), db/db mice (Kinney Int. 2013; 84: 545. PMID: 23615507), and high-fat diet (HFD)-fed mice (Int J Obes. 2012; 36: 1062, PMID: 22184057).

With regarding to ROCK1, we performed western blot and qPCR analysis in animal kidney samples. As shown in Supplementary Fig. 1b and c, protein and mRNA expression of ROCK1 was significantly increased in STZ-injected mice, and there was an increasing trend in the kidney of db/db mice and in HFD-fed models. These results suggest that Rho/ROCK pathway is activated in the kidney of diabetic animals. We have included these data and references in the text on page 4 starting at line 75 as follows,

..... The ROCK2 protein levels were also elevated in db/db mice and high-fat diet (HFD)-fed mice, both of these are common type 2 diabetes models for studies of diabetic kidney disease. **In these models, RhoA (Supplementary Fig. 1a) and the downstream target myosin phosphatase targeting subunit 1 (MYPT1) are activated^{3,9,10}. Moreover, there was a trend towards elevations in renal ROCK1 levels in diabetic animals (Supplementary Fig. 1b and c).** Glomeruli isolated from these models exhibited a significant transcription-level increase in ROCK2 in comparison to glomeruli obtained from non-diabetic mice (Fig. 1b).

Methods section and figure references were edited appropriately.

Comment 2 - In figure 1c and 5j, isn't R2 low to say that two variants are closely related each other?

Our response – We understand the reviewer's concern. The coefficient of determination, R^2 , was 0.16 in Fig. 1c. $R^2 = 0.16$ in this figure means that 16% of the total variation in ROCK2 mRNA levels can be explained by the linear relationship between ROCK2 mRNA and albumin to creatinine ratio (ACR). The other 84% of the total variation in ACR remains unexplained. Therefore, the strength of the linear association between ROCK2 mRNA levels and ACR is not strong as the reviewer suggested. Similarly, $R^2 = 0.23$ in Fig. 5j means that ROCK2 mRNA levels and PPAR α mRNA levels are not closely related each other. We have edited our manuscript as follows (page 4 at line 82 and page 7 at line 156),

..... Importantly, transcript datasets obtained from Nephroseq (<https://www.nephroseq.org>) showed a **mild** positive correlation between the glomerular expression of ROCK2 and urinary albumin excretion, measured as the albumin-to-creatinine ratio (ACR), in murine models of diabetes (Fig. 1c), **that may suggest** the association of activated ROCK2 signaling with albuminuria.

..... Consistently, glomerular PPAR α was upregulated in PR2KO mice (Fig. 5i) and there was a **mild** negative correlation between glomerular ROCK2 and the expression of PPAR α (Fig. 5j), further indicating a role of ROCK2 as a **potential** negative regulator of PPAR α

Comment 3 - In figure 1d, the image seems not well focused. It seemed that the images were taken at different depth when considering the morphology of nucleus. Depending on the depth, the brightness will be measured differently. Also, how many fields were taken and tested?

Our response – Thank you for pointing that out. In Fig. 1d, we have stained two serial sections because antibodies against both ROCK2 and nephrin were raised in rabbit. We therefore changed nephrin antibody to guinea pig in order to evaluate distribution and expression levels of ROCK2 and nephrin at the same depth. As shown in revised Fig. 1d, high resolution double immunostaining images taken at the same depth has been added. Twenty glomeruli were observed per section using imaging system. We have edited the methods section to clarify this (page 11 starting at line 257).

..... The following antibodies were used for staining: anti-ROCK2 (abcam #ab71598), anti-Nephrin (abcam #ab227806, ORIGENE #1243-1256), and anti-WT1 (abcam ab89901). **For**

double immunostaining with ROCK2 and nephrin in mice, 20 glomeruli were observed per section and representative images taken at the same depth are demonstrated. To determine the loss of glomerular podocytes, the number of WT1-positive cells was counted in 20 glomeruli per section.

Comment 4 - How many animals and images were examined in figure 3? To examine the significance, more tests will be needed.

Our response – In imaging analysis, 3 to 6 mice per group were examined for the evaluation of glomerular sclerosis. The number of mice used for immunostaining for WT1 and EM analysis was 3 per group. In the revised manuscript, we have performed these experiments using at least 4 mice per group including db/db study. As demonstrated in Fig. 3, the progression of glomerular sclerosis, podocyte loss, foot process and GBM abnormalities were attenuated in three kinds of diabetes models (i.e., STZ injection, db/db, HFD feeding). We have edited figures and figure legends as follows,

Fig. 3: Ablation of podocyte ROCK2 prevents diabetic renal damage in mice. **a**, The breeding scheme that was used to generate PR2KO mice. Diabetes was induced by STZ injection, mating with db/m mice to generate db/db mice, or HFD feeding. **b**, ACR in STZ-injected, db/db, HFD-fed WT and PR2KO mice (n = 4-8). **c**, The kidney weight in the three experimental groups (n = 4-8). **d**, Representative PAS-stained images of kidney glomeruli from mice. The scale bar on the top left represents 10 μm (n = 4-6). **e**, WT1 immunostaining and quantification of WT1-positive cells in glomeruli from mice. The scale bar on the top left represents 10 μm (n = 4). **f**, Foot process width examined by transmission electron microscopy. The scale bar on the top left represents 1 μm (n = 4). **g**, GBM thickness assessed by transmission electron microscopy. The scale bar on the top left represents 0.5 μm (n = 4). *p < 0.05. Data represent the mean \pm s.e.m.

Comment 5 - In RNA-sequencing analysis, supportive experiments are needed, for example qRT-PCR analysis for fatty acid metabolism-related genes.

Our response – Thank you for the suggestion. We have performed qPCR analysis in order to confirm the data obtained from RNA-Seq. As shown in revised Supplementary Fig. 3, qPCR analysis demonstrated that core essential genes of fatty acid metabolism, including Cpt1b, Cpt2, Acox2, Ppara, were significantly upregulated in ROCK2-knockdown podocytes. In addition, upregulation of these genes was observed in podocytes treated with ROCK2 inhibitor KD025. These data support the results obtained from RNA-seq analysis. We have edited manuscript in order to include these data as follows (page 6 starting at line 120),

..... As shown in Fig. 4d, a KEGG pathway enrichment analysis revealed significant changes in the metabolic pathway. **Upregulation of core essential genes in fatty acid metabolism, including Cpt1b, Cpt2, Acox2, Ppara, were confirmed by qPCR analysis in podocytes treated with ROCK2 siRNA (Supplemental Fig. 3c) or ROCK2 inhibitor (Supplemental Fig. 3d). Among these genes, Cpt1b, Acox2 (Supplementary Fig. 3e), and Ppara (Fig. 5i) were significantly upregulated in glomeruli isolated from PR2KO mice compared to WT mice. These data indicate that ROCK2 is involved in essential pathway of fatty acid oxidation (FAO) in podocytes.**

Primer sequences that were used for the qPCR have been added to Supplementary Table 2.

Comment 6 - Authors showed the effect of ROCK2-deficiency on TGF β in podocyte in Figure 5. It seems that ROCK2 downregulate TGF β in the results. What is the link between ROCK2 and TGF β expression? Also, what is the effects of ROCK2 deficiency on fatty acid metabolism in cells and animals?

Our response – As the reviewer pointed out, it is true that ROCK is implicated in TGF- β expression in certain cell types. For example, ATP competitive ROCK inhibitors, Y-27632 and fasudil, have been shown to inhibit TGF- β production induced by advanced glycation end products in human peritoneal mesothelial cells (PMID: 29581852). In our additional experiments, however, TGF- β mRNA levels were not changed in ROCK2-deficient podocytes (Supplemental Fig. 4f). In addition, ROCK2 inhibitor did not downregulate TGF- β expression in podocytes (Supplemental Fig. 4g). Therefore, it seems that TGF- β is nor regulated by ROCK2 at least in podocytes. The relationship between ROCK and TGF- β may vary depending on cell types and culture conditions. We have included these data and edited manuscript on page 7 starting at line 151 as follows,

..... The upregulation of PPAR α and FAO, and the prevention of podocyte apoptosis were also observed with the pharmacological blockade of ROCK2 (Supplemental Fig. 4b-e). **The expression levels of TGF- β were not changed in ROCK2-dificent conditions (Supplemental Fig. 4f) and in podocytes treated with ROCK2 inhibitor (Supplemental Fig. 4g).** Consistently, glomerular PPAR α was upregulated in PR2KO mice (Fig. 5i) and there was a mild negative correlation between glomerular ROCK2 and the expression of PPAR α (Fig. 5i), further indicating a role of ROCK2 as a potential negative regulator of PPAR α

As shown in Fig. 5e, ROCK2 deficiency improves TGF- β -induced downregulation of fatty acid oxidation. qPCR analysis confirmed induction of genes involved in fatty acid metabolism in these cells (Fig. 5f). In the revised manuscript, besides these mediators, data showing upregulation of Cpt1b, Cpt2, Acox2, Ppara genes in ROCK2-deficient cells and in ROCK2 inhibitor-treated podocytes has been added (Supplemental Fig. 3c and d). Among these, Cpt1b, Acox2

(Supplementary Fig. 3e), and Ppara (Fig. 5i) were significantly upregulated in glomeruli isolated from podocyte-specific ROCK2 knockout mice compared to wild-type mice. We edited manuscript as follows (page 6 starting at line 120),

..... As shown in Fig. 4d, a KEGG pathway enrichment analysis revealed significant changes in the metabolic pathway. **Upregulation of core essential genes in fatty acid metabolism, including Cpt1b, Cpt2, Acox2, Ppara, were confirmed by qPCR analysis in podocytes treated with ROCK2 siRNA (Supplemental Fig. 3c) or ROCK2 inhibitor (Supplemental Fig. 3d). Among these genes, Cpt1b, Acox2 (Supplementary Fig. 3e), and Ppara (Fig. 5i) were significantly upregulated in glomeruli isolated from PR2KO mice compared to WT mice. These data indicate that ROCK2 is involved in essential pathway of fatty acid oxidation (FAO) in podocytes.**

Comment 7 - Was podocyte cell death detected in disease models, and the recovery from ROCK2-deficient mice?

Our response – Thank you for pointing that out. We performed TUNEL assay in order to assess podocyte cell death in glomeruli. As demonstrated in Supplementary Fig. 2, we found that the number of apoptotic cells including peripheral area of glomeruli was significantly increased in disease models, which was decreased in podocyte-specific ROCK2-deficient mice. Our data indicate that ROCK2 deficiency inhibits apoptosis of glomerular cells. This effect was not restricted to podocytes in glomerulus. We have edited manuscript as follows (page 5 starting at line 106),

..... While significant disruption in Wilms tumor 1 (WT1) was observed in diabetic mice, PR2KO mice were protected from podocytes loss (Fig. 3e). **TUNEL-assay showed that diabetes-induced glomerular cell apoptosis was decreased in PR2KO mice (Supplementary Fig. 2c).** As shown in Fig. 3f and g, transmission electron microscopy (TEM) micrographs from PR2KO mice revealed a significant reduction in foot process width and glomerular basement membrane (GBM) thickness in comparison to ROCK2-floxed mice in the setting of diabetes.

Methods section was edited appropriately.

Reviewer #2: *In the manuscript entitled “ROCK2-induced metabolic rewiring in diabetic podocytopahty” submitted by Matoba et al, the authors investigated the role of podocyte ROCK2 in response to diabetic injury. They reported that podocyte ROCK2 is activated in 3 different rodent diabetic models and in patients with diabetes. They found that selective deletion of ROCK2 in podocytes has no effect on development but protects against diabetic kidney injury in different diabetic models. They further found that the beneficial effect of ROCK2 inhibition is due to rescue of PPAR α signaling, leading to recovery of*

podocyte fatty acid metabolism. This is a well-designed and well-performed study with clear results and presentation. A minor issue is that high resolution double IF photos are needed.

Our response – We thank the referee for these insightful comments and constructive suggestions. High resolution double immunostaining has been added to Fig. 1.

Reviewer #3: *In this report, Matoba and colleagues demonstrate a novel role for ROCK2 in podocytes in mediating diabetic renal complications. They do so using a combination of a novel transgenic animal model lacking ROCK2 in podocytes (PR2KO) and immortalized podocytes treated in culture with anti ROCK2 siRNA. Inhibition of ROCK2 limits renal injury in three different models of diabetes. They also demonstrate a role for ROCK2 in modulating PPAR α expression and fatty acid oxidation in podocytes. Overall, the studies are well-described and carried out, but certain conclusions may not be fully supported or could be further strengthened.*

Our response - We appreciate the reviewer for kind review and valuable comments that helped us in improving our manuscript.

Comment 1 - Methods: For db/db model, why were db/+ ROCK2^{flox/flox} mice used as controls rather than db/db ROCK2^{flox/flox}? db/+ are not suitable controls in this scenario.

Our response – We apologize for any confusion related to controls used in db/db study. In Fig. 3, we confirmed the progression of functional and histological abnormalities in diabetic db/db ROCK2^{flox/flox} animals by comparing with db/m ROCK2^{flox/flox} mice. In order to evaluate the effects of podocyte ROCK2 deletion on the diabetic nephropathy, we compared db/db ROCK2^{flox/flox} mice with db/db ROCK2^{flox/flox} mice in four groups (i.e., db/m ROCK2^{flox/flox}, db/m PR2KO, db/db ROCK2^{flox/flox}, db/db PR2KO). We have deleted the term “controls” to prevent confusion for the reader. Methods section on page 10 starting at line 221 was edited as follows,

..... Diabetic db/db mice and their **non-diabetic** littermates, db/m, were obtained from The Jackson Laboratory. Since db/db mice are infertile, db/db PR2KO mice were bred by mating db/m ROCK2^{flox/flox} mice with db/m PR2KO mice. Lean controls were db/m ROCK2^{flox/flox} animals.

Comment 2 - Figure 4 - what level of ROCK2 inhibition was achieved? What does siCtrl refer to - a control gene or a scrambled control? How much lipofectamine was used and for how long?

Our response – We thank the referee for raising this important point. In qPCR analysis, ROCK2 mRNA levels were 36.6% in cells treated with siRNA against ROCK2 compared to podocytes treated with scrambled siRNA (Supplemental Fig. 3a). ROCK2 protein expression was 29.9% in

ROCK2-knockdown cells compared to scrambled siRNA-treated cells (Supplemental Fig. 3b). We used control siRNA which consists of a scrambled sequence that will not lead to the specific degradation of any cellular message. Podocytes were transfected with siRNAs using 1 μ L of Lipofectamine reagent. Growth medium was added after 6 hours. We have edited methods section as follows (page 10 starting at line 239),

ROCK2 knockdown podocytes were established by incubating cells with transfection mixtures containing 50 nM of siRNA. A scrambled sequence that will not lead to the specific degradation was used as control. Transfection of siRNAs with 1 μ L of Lipofectamine reagent (Invitrogen, Carlsbad, CA, USA) for 6 hours resulted in approximately 63% mRNA knockdown and 71% knockdown at protein levels (Supplemental Fig. 3b). Twenty-four hours after adding growth medium, the transfected podocytes were stimulated with TGF- β (5 ng/mL) for 24 hours.

Comment 3 - In Figure 4, transcriptomics were carried out in immortalized podocytes rather than in primary cells. While this is experimentally justified, this information should be more clearly conveyed in the text accompanying Figure 4 (i.e. line 111, not only in the methods section). Further, it should be clarified in the text that transcriptional differences observed are following stimulation by TGF-b, and the rationale for the use of TGF-b should be clarified here, rather than later in the manuscript. On a related note, was a baseline (non-TGFb treated) transcriptomics experiment carried out? Fig. 4B: Label as siR2+TGFb, as in Fig. 5g.

Our response – Thank you for pointing that out. To clarify this and avoid potential confusion to readers, we have edited results section as follows (page 6, line 116),

We then explored the molecular mechanisms by which ROCK2 deficiency exerts protective effects in podocytes. We deleted ROCK2 in immortalized podocytes and characterized the transcriptomic profiles by performing unbiased RNA-sequencing (Fig. 4a). An analysis of all expressed genes revealed 298 significantly downregulated and 306 significantly upregulated genes (fold change > 1.5; p value < 0.05) (Fig. 4b and c).

In RNA-Seq analysis, the transcriptomic changes were observed in non-stimulated podocytes. We apologize for any confusion resulting from the statements in our article. We have edited heading of the result section as follows (page 5 at line 114),

ROCK2 deletion results in robust metabolic signature in non-stimulated podocytes. We then explored the molecular mechanisms by which ROCK2 deficiency exerts protective effects in podocytes.

Comment 6 - In Figure 5F, while authors show that treatment of podocytes with siR2 prevents apoptosis, they do not demonstrate that this is dependent on fatty acid oxidation. In other words, if these cells were also treated with etomoxir, would the protective effect of siR2 be lost? Also, if these effects in FAO are mediated by PPAR α , does treatment with fibrates induce the same protective effect as ROCK2 inhibition? And would co-treatment with siR2 and fibrates result in additive or no additive effects? These crucial experiments are necessary to conclude that the cytoprotective effects of ROCK2 inhibition are due to enhanced FAO and that this enhancement of FAO is due to PPAR α activation. This is especially true given the reported changes in AMPK signaling (Fig. 4), another mechanism through which FAO may be regulated.

As shown here, the data demonstrate that ROCK2 inhibition is protective in the context of diabetes-induced renal injury and that ROCK2 inhibition is ASSOCIATED with rescue of FAO and PPAR α expression. However, the data as shown do not support the conclusion that the cytoprotective effects of ROCK2 inhibition are mediated by PPAR α or by induction of FAO. These endpoints could just as easily be a consequence of other changes occurring due to ROCK2 inhibition.

Our response – The reviewer is absolutely right. We thank the reviewer for raising the important issue on the effects of ROCK2 deletion. Based on the reviewer's suggestion, we have conducted experiments in order to investigate if siROCK2-mediated cytoprotective effects are dependent on fatty acid oxidation. As shown in revised Fig. 5h, the protective action of ROCK2 silencing was significantly attenuated by the treatment of etomoxir. However, this was partial cancellation, indicating that siROCK2-mediated protective action is not completely dependent on fatty acid oxidation. Other biological pathways may also be regulated by ROCK2 inhibition.

Fenofibrate was used to assess the effect of PPAR α activation on podocyte apoptosis. Importantly, the number of TUNEL-positive cells was decreased by the treatment of fenofibrate. Additive effects of ROCK2 siRNA and fenofibrate were also observed. These data support the idea that the beneficial actions of ROCK2 deficiency is dependent PPAR α activation. We have edited manuscript as follows (page 7 starting at line 144),

..... We reasoned that one possible explanation for this action could be through the effect of ROCK2 on PPAR α , since the key FAO enzymes were upregulated by PPAR α agonist in podocytes (Supplemental Fig. 4a) and the genetic ablation of ROCK2 was associated with recovery from the PPAR α suppression induced by TGF- β (Fig. 5g). **As demonstrated in Fig. 5h, TGF- β -induced podocyte apoptosis was inhibited in the setting of ROCK2 deletion. This beneficial action was partially canceled by the treatment with etomoxir, indicating that cytoprotective effects of ROCK2 inhibition is dependent, at least in part, on fatty acid oxidation. Treatment with fenofibrate showed similar protective effect as ROCK2 deletion in podocytes, and co-treatment with fenofibrate and ROCK2 siRNA induced additive effects on podocyte protection. These data support the idea that the beneficial actions of ROCK2 deficiency is**

dependent on PPAR α activation. The upregulation of PPAR α and FAO, and the prevention of podocyte apoptosis were also observed with the pharmacological blockade of ROCK2 (Supplemental Fig. 2b-e).

REVIEWERS' COMMENTS:

Reviewer #1 (Remarks to the Author):

Dear authors;

Every comments were amazingly well solved and the manuscript quality was much improved. This version is well documented for the publication in Communications Biology.

Reviewer #3 (Remarks to the Author):

The authors have provided thoughtful and appropriate responses to concerns raised. This report is now recommended for acceptance and publication.